# *SLC4A11* Revisited: Isoforms, Expression, Functions, and Unresolved Questions

**DOI:** 10.3390/biom15060875

**Published:** 2025-06-16

**Authors:** Polina Alekseevna Kovaleva, Elena Sergeevna Kotova, Elena Ivanovna Sharova, Liubov Olegovna Skorodumova

**Affiliations:** Medical Genomics Laboratory, Lopukhin Federal Research and Clinical Center of Physical-Chemical Medicine of the Federal Medical Biological Agency, 119435 Moscow, Russia; polilukoe@gmail.com (P.A.K.); potamanthus@gmail.com (E.S.K.); sharova78@gmail.com (E.I.S.)

**Keywords:** SLC4A11, congenital hereditary endothelial dystrophy, Harboyan syndrome, cornea, corneal endothelium, Fuchs endothelial corneal dystrophy, variants, cancer, lactate, glutamine, ammonia

## Abstract

The *SLC4A11* gene encodes a membrane transporter implicated in congenital hereditary endothelial dystrophy, Harboyan syndrome, and certain cancers. Despite its clinical importance, current data on *SLC4A11* expression patterns, transcript variants, and functional roles remain inconsistent and sometimes contradictory. We have systematized existing data, identified areas of consensus, and highlighted discrepancies. This review addresses *SLC4A11* transcript and isoform diversity and how this complexity influences both the interpretation of its tissue expression patterns (particularly in the corneal endothelium) and the investigation of its functional roles in health and disease. Our review also untangles the evolving understanding of SLC4A11 function, from its initial classification as a bicarbonate transporter to its established roles in NH_3_- and pH-regulated H^+^/OH^−^ transport, lactate efflux, cellular stress responses, and adhesion. The review details how pathogenic mutations disrupt protein maturation, membrane localization, or transport activity, contributing to corneal fluid imbalance and disease. We also discuss the emerging role of SLC4A11 in cancer metabolism and the common metabolic features of dystrophic corneas and tumors. Methodological challenges are appraised, encouraging caution in interpretation and the need for isoform-specific studies. Overall, this review provides a comprehensive update on SLC4A11 biology and identifies key gaps for future research.

## 1. Introduction

The solute carrier family 4 member 11 (*SLC4A11*) gene—also known as bicarbonate transporter-related protein-1 (BTR1) or Na+-coupled borate cotransporter 1 (NaBC1)—plays a significant role in the pathogenesis of several diseases [1,2]. Homozygous and compound heterozygous variants in *SLC4A11* cause congenital hereditary endothelial dystrophy (CHED, formerly CHED2) [3,4] and Harboyan syndrome, which combines corneal dystrophy with sensorineural deafness [5,6,7]. Rare *SLC4A11* variants have also been linked to Fuchs corneal endothelial dystrophy (FECD) [8,9], though the evidence for a causal role in FECD remains insufficient [10]. Recent studies report elevated SLC4A11 protein levels in certain cancers, suggesting a potential role in tumor growth, particularly in glutamine-dependent cancers that rely on glutamine for proliferation [11,12].

Reports on the tissue specificity of *SLC4A11* expression are inconsistent. Some studies describe *SLC4A11* as ubiquitously expressed [2,12], while others do not detect its mRNA or protein in all tissues [1,13]. The fact that *SLC4A11* mutations cause diseases restricted to the cornea and inner ear, without broader syndromic features characteristic of ubiquitous genes, indirectly supports tissue-specific expression.

There is also no consensus on the predominant transcript variant or protein isoform of SLC4A11. Transcript variant 2 was initially identified and long considered to be the predominant transcript variant, with its protein product (isoform 2) thought to be a major isoform of the SLC4A11 protein [1]. However, subsequent studies have shown that in the corneal endothelium, the tissue most affected by *SLC4A11*-related diseases, the major protein is an N-terminal shortened isoform [14]. Moreover, it was shown that another transcript variant (variant 3) is expressed in the corneal endothelium at similar or even higher levels than variant 2 [15].

The understanding of SLC4A11 function has also evolved. Originally identified due to sequence similarity with bicarbonate transporters [1], SLC4A11 was initially hypothesized to transport bicarbonate. It was also hypothesized that SLC4A11 transports sodium cations and borate ions simultaneously [2]. However, subsequent studies have shown that it does not transport bicarbonate or borate [16,17]. Conflicting evidence exists regarding its ability to transport Na^+^, NH4^+^, OH^−^, or H^+^ ions. For each small molecule or ion, studies have been found both supporting and refuting these roles [16,17,18,19,20,21]. Additionally, SLC4A11 has been shown to function as a mitochondrial proton uncoupler [22] and to be involved in cell–cell contact structures [23,24].

Given the established association of *SLC4A11* mutations with CHED and Harboyan syndrome, and its potential involvement in glutamine-dependent cancers, investigating SLC4A11’s role in disease pathogenesis is important for medical genetics. However, significant challenges remain in reconciling data on tissue specificity, protein isoforms, and functional features. This review aims to provide a structured and comprehensive summary of current knowledge on SLC4A11, synthesizing findings across different tissues and species.

## 2. SLC4A11 Gene Expression, Resulting Protein Isoforms, and Cellular Localization

### 2.1. Transcript Variants and Isoforms of the Protein

Research on the *SLC4A11* gene began when its cDNA sequence was identified in the GeneBank by searching for genes homologous to known bicarbonate transporters [1]. The described sequence corresponded to transcript variant 2 of the *SLC4A11* gene (Table 1, Figure 1A). The 5′-end of this transcript was confirmed to match the GeneBank sequence using 5′-RACE with a commercial kidney cDNA sample (Kidney Marathon Ready cDNA, Clontech). The full-length transcript was then cloned into a vector using primers designed with a modified, closer-to-consensus (strong) Kozak sequence at the first ATG codon. The protein product was expressed in an *in vitro* system, glycosylation was shown to be possible, and electrophoretic mobility was characterized [1].

**Table 1 biomolecules-15-00875-t001:** Match table of the main isoforms and transcript variants of the *SLC4A11* gene.

Isoform Names [14,25]	Isoform Length (aar)	RefSeq Isoform Name and Protein ID in Consistence with Publications [14,25]	RefSeq mRNA Name and ID in Consistence with Publications [14,25]	Other RefSeq SLC4A11 Isoforms with 100% Amino Acid Sequence Identity (Isoform Name: Protein ID/mRNA ID)	Tissue, from Which the Corresponding Transcript Was Isolated	First Mention
SLC4A11-A, SLC4A11-v1	918	isoform 1 (NP_ 001167561)	transcript variant 1 (NM_ 001174090)	-	brain	[25]
SLC4A11-B, SLC4A11-v2-M1	891	isoform 2 (NP_ 114423)	transcript variant 2 (NM_032034)	isoform X2: XP_047296496/ XM_047440540	kidney	[1]
SLC4A11-v2-M36	856	-	transcript variant 2 (NM_032034)	isoform 5: NP_001387206/ NM_001400277, NP_001387207/ NM_001400278, NP_001387208/ NM_001400279, isoform X4: XP_016883585/ XM_017028096, XP_016883583/ XM_017028094	corneal endothelial cells	[14]
SLC4A11-C, SLC4A11-v3	875	isoform 3 (NP_ 001167560)	transcript variant 3 (NM_ 001174089)	-	corneal endothelial cells	[25]

The first publication to mention other SLC4A11 isoforms (Table 1) was by Kao et al. in 2015 [25]. These isoforms had apparently already been described in the NCBI RefSeq database [26], as referenced by Kao et al. with specific RefSeq identifiers. We found no earlier references to these isoforms, and subsequent authors also cite Kao et al. when discussing them [19]. Full-length cDNAs corresponding to the transcript variants were amplified from cDNA and RNA samples from various tissues (Table 1), and the resulting protein isoforms were characterized. The transcript variants differed in the 5′-terminal region, which included a small portion of the reading frame. The corresponding protein isoforms differ by a small N-terminal sequence and have little difference in length (Table 1). Isoforms 2 and 3 were localized to the plasma membrane, whereas isoform 1 was found in the cytoplasm. In this and subsequent studies by the same group, variant 3 was shown to be the major variant in the corneal endothelium [15,25].

**Figure 1 biomolecules-15-00875-f001:**
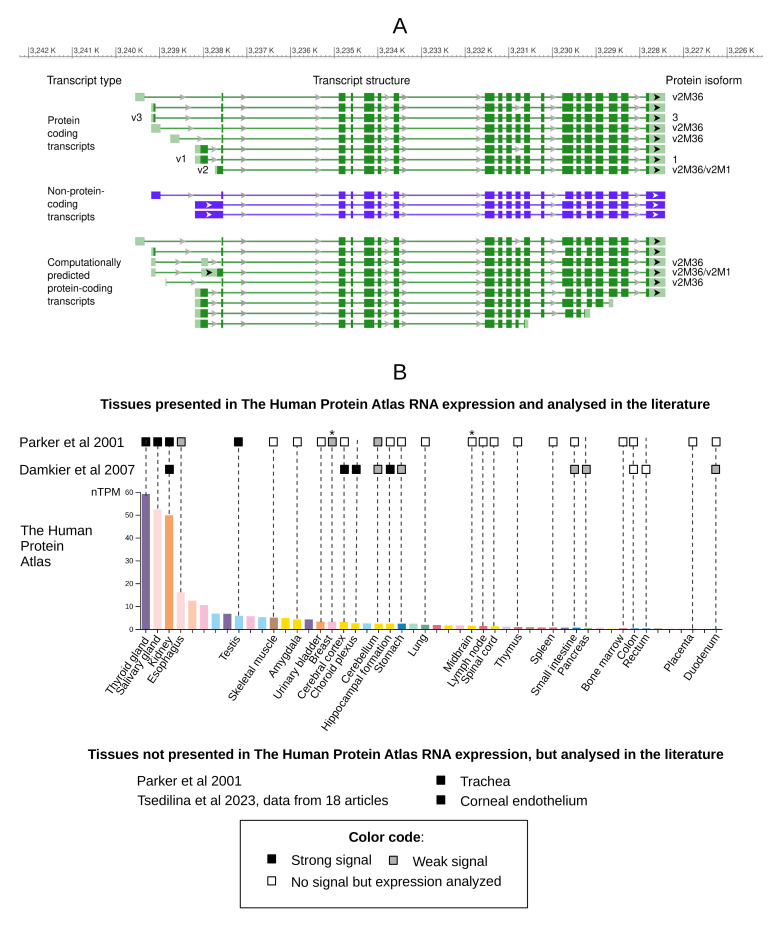
*SLC4A11* gene expression. (**A**). Transcript variants of *SLC4A11* presented in RefSeq database (GRCh38.p14 genome assembly, image obtained 12 April 2025). The boxes correspond to exons, the lines connecting them correspond to introns, and the arrows indicate the direction of transcription. (**B**). Schematic visualization of *SLC4A11* expression in normal human tissues. Comparison of data obtained from The Human Protein Atlas [27] and from two publications that analyzed *SLC4A11* expression in tissue sets provided at the top of (**B**). Tissues that express *SLC4A11* mRNA but not are presented in The Human Protein Atlas are presented at the bottom of (**B**). *SLC4A11* expression data were arranged in descending order. nTPM—normalized transcripts per million in consensus The Human Protein Atlas dataset. * Tissues from the publication of Parker et al. that do not exactly correspond to tissues in The Human Protein Atlas: mammary gland in article and breast in The Human Protein Atlas; thalamus in article and midbrain in The Human Protein Atlas. Publications: Parker et al., 2001, Damkier et al., 2007, and Tsedilina et al., 2023 correspond to references with numbers [1,10,13], respectively.

However, a 2019 study by Malhotra et al. [14] using quantitative PCR on a cDNA matrix found contrasting results to those of Kao et al. in 2015 [25]: transcript variant 2 was expressed slightly higher than variant 3 in the corneal endothelium [14]. Malhotra et al. [14] also demonstrated that the corneal endothelium preferentially produces a protein from transcript variant 2, not from the first ATG codon with a “weak” Kozak sequence (SLC4A11-B or v2M1), but from a second ATG codon in the same reading frame, that has a stronger, closer-to-consensus Kozak sequence. This isoform, named v2M36, starts at the 36th methionine residue of the v2M1 isoform. The level of v2M36 in corneal endothelial lysates was found to be four times higher than that of the SLC4A11-C isoform encoded by transcript variant 3 [14]. Both v2M36 and v2M1 isoforms are localized to the plasma membrane.

Interestingly, while previous authors based their statements on the existence of three RNA variants of *SLC4A11*, the current version of the RefSeq database [26] lists 20 transcript variants (Figure 1A): 8 are known protein-coding transcripts, 9 are predicted protein-coding transcripts, and three are known non-coding transcripts. In addition to transcript variant 2, there are other known and predicted protein-coding transcripts whose reading frame corresponds to v2M36 (Table 1). Moreover, transcript variants 1–3 do not contain completely unique regions for primer design, making it difficult to specifically detect each variant without cross-reactivity with other described *SLC4A11* transcript variants. As a result, PCR analyses targeting these variants may also detect other transcripts, potentially confounding expression results.

All protein-coding transcript variants of *SLC4A11* listed in RefSeq, which include sequences encoding the full-length transmembrane domain, correspond to protein isoforms SLC4A11-v1, SLC4A11-v2/SLC4A11-v2M36, SLC4A11-v2M36, and SLC4A11-v3. However, RefSeq also contains transcripts that lack the exon corresponding to exon 10 of NM_032034 (e.g., NM_001363745, NM_001400280, XM_047440543), which are the result of alternative splicing. Exon 10 encodes the second transmembrane region and most of the third transmembrane region. Its absence produces protein products missing these segments.

Additionally, some predicted transcripts include truncated exons 13 and 15 (numbered according to NM_032034). These alterations affect extracellular loops that are part of the transmembrane domain. Truncation of exon 13 removes part of the large extracellular loop 3 (EL3), which contains N-glycosylation sites. However, the sites themselves remain intact. Truncation of exon 15 results in the loss of most of extracellular loop 4 (EL4). It is noteworthy that these splicing events do not result in frameshifts, indicating that the resulting proteins are canonical isoforms with deletions in their polypeptide chains. The inclusion of truncated exons 13 and 15 does not disrupt transmembrane regions but shortens intervening polypeptide segments. Deletions in EL3 and EL4 may either preserve normal protein function or severely compromise it, particularly if such deletions affect the structural organisation of the transmembrane domain (e.g., proper folding or spatial orientation of transmembrane regions).

RefSeq also lists potential transcripts encoding C-terminally truncated SLC4A11 isoforms, a consequence of premature transcription termination. These variants would lack the C-terminal domain and part of the transmembrane domain (up to all transmembrane regions except the first). Such isoforms could differ markedly from canonical full-length transmembrane domain isoforms in subcellular localization, maturation, and function.

The functional role of SLC4A11 isoforms with polypeptide deletions or truncations (via alternative splicing or premature termination) remains unexplored. Priority should be given to analyzing the prevalence of these splice variants. They may lack functional significance, representing low-abundance stochastic splicing byproducts. However, if produced at substantial levels, functional assays could elucidate their cellular roles.

Our recent work addressed the above-mentioned gaps and contradictions regarding the expression of *SLC4A11* protein-coding transcripts [28]. Using transcriptomic analysis, we revealed the predominant expression of protein-coding transcript variants 6 and 3, but not variants 2 and 1, in the corneal endothelium of patients with FECD and controls.

In addition, the *SLC4A11* promoter region was identified and characterized by bioinformatic analysis [29,30]. The promoter regions identified in these studies largely overlap, but notably do not include the transcription initiator for variant 2. The proposed promoter region was located 1 kb upstream of the variant 2 transcription start site and included the transcription start site for variant 3. This controversy was resolved in our recent work, where we used the 5′ RACE method to directly show that the major transcription start site is located at the beginning of transcript 6, from which variant 3 with an extended 5′ UTR is also transcribed [28].

In summary, the regulation of *SLC4A11* gene expression and the relative abundance of its transcript variants remain incompletely understood and warrant further investigation.

### 2.2. SLC4A11 mRNA Detection

Data from the literature on the tissue specificity of *SLC4A11* expression show some inconsistencies. Some studies report that *SLC4A11* is ubiquitously expressed [2,12]. However, other studies using reverse transcription polymerase chain reaction (RT-PCR) have failed to detect *SLC4A11* transcription in certain tissues, and in other tissues *SLC4A11* was expressed at varying levels [1,13].

In the first study describing the *SLC4A11* (BTR1) gene, Parker et al. analyzed transcript expression by Northern blot with a chip containing RNA samples from numerous human tissues, including the brain, spinal cord, heart, and gastrointestinal tract [1]. They detected *SLC4A11* RNA in the kidney, trachea, thyroid, salivary glands, and testis [1]. Damkier et al. analyzed *SLC4A11* mRNA by RT-PCR in 13 human tissues. *SLC4A11* expression was detected in the renal cortex and medulla, stomach, ileum, duodenum, pancreatic body, choroid plexus, hippocampus, cerebellum, and cerebral cortex [13]. The strongest expression was in the renal medulla, with reliable detection in the renal cortex. Signals in the stomach, ileum, pancreatic body, and cerebellum were weak, indicating low transcript levels, while no expression was detected in the pancreatic head, jejunum, or colon [13].

Both studies agree on high *SLC4A11* expression in the kidney, weak expression in the cerebellum, and absence in the colon [1,13]. However, they differ regarding other brain regions (cerebral cortex, hippocampus) and digestive tract tissues (small intestine, duodenum): Parker et al. [1] found no expression, whereas Damkier et al. [13] detected signals of variable intensity.

These findings align broadly with data from the Human Protein Atlas (version 24.0) [27,31] (Figure 1B), which show the highest *SLC4A11* expression in the thyroid, salivary gland, kidney, esophagus, and skin, and minimal expression in the liver, ovary, placenta, tongue, and duodenum. Notably, tissues with weak expression in the Human Protein Atlas correspond to those where Parker et al. [1] reported no expression but Damkier et al. [13] detected weak signals, likely reflecting differences in method sensitivity. Additionally, Parker et al. [1] reported *SLC4A11* expression in tracheal tissue, a tissue which is absent from the Human Protein Atlas [27,31] and Damkier et al.’s work [13].

The expression of *SLC4A11* is well-documented in corneal tissues. At least 18 studies have reported high expression levels in the corneal endothelium [10], with much lower expression in the corneal stroma and near absence in the multilayered anterior epithelium [28,32].

Since *SLC4A11*-related inherited diseases primarily affect the corneal endothelium, its expression there has been extensively studied. However, we found no studies except one that directly compared *SLC4A11* expression in the corneal endothelium with that in non-corneal tissues [28]. This gap likely reflects the challenges of working with corneal tissues, especially the endothelium, and the consequent lack of corneal samples in tissue arrays and panels such as those used by Parker et al. and in the GTEx project samples [1,13,33].

### 2.3. SLC4A11 Protein Detection

Immunohistochemical (IHC) staining of the human renal medulla revealed SLC4A11 localization primarily in the plasma membrane, especially on the basolateral side, of collecting duct epithelial cells. SLC4A11 was also detected in the membranes of renal podocytes and in the brush border of proximal tubules in the renal cortex. Additionally, staining was observed in the renal cortex and outer medulla, corresponding to the apical membrane domains of intercalated cells. A similar distribution has been reported in rat kidney tissue [13]. IHC analysis of mouse and rat kidneys showed that SLC4A11 staining overlapped with aquaporin 1 (AQP1) in the same structures [34]. Both proteins were detected in the upper part of the thin descending limb of the loop of Henle in long-looped nephrons. Colocalization of SLC4A11 with AQP1 and the urea transporter UT-B was also observed in the mouse descending vasa recta, although SLC4A11 was not detected in the rat vasa recta. Both SLC4A11 and AQP1 staining intensity decreased from the inner renal medulla towards the cortex, where their presence was almost absent [34]. These findings align with Damkier et al.’s mRNA expression data [13]. The staining of SLC4A11 in some structures of the human and rat renal cortex in Damkier et al., but the absence of signal in the mouse renal cortex in Gee et al., may be due to differences in the sensitivity of IHC in the two papers [13,34].

In human pancreatic tissue, IHC revealed SLC4A11 predominantly in the intercalated ducts, especially on membranes facing the duct lumen. In the brain, SLC4A11 localization was restricted to the apical membrane and brush border of choroid plexus epithelial cells and capillary endothelium [13].

In the cornea, SLC4A11 was detected in both the anterior multilayer epithelium and the posterior monolayer epithelium (more commonly referred to as the corneal endothelium). Early studies reported predominantly intracellular localization in the corneal endothelium [13], whereas more recent work demonstrated membrane localization, mainly on basolateral membranes in human [35,36], mouse [37], and rabbit corneal endothelium [17]. SLC4A11 was also found in the mitochondria of human corneal endothelial cells by immunofluorescence (IF) and Western blot, and in mouse by Western blot [22], although later IF studies showed only weak co-localization with mitochondrial markers [36].

Due to progressive hearing loss in carriers of pathogenic homozygous or compound heterozygous SLC4A11 variants with Harboyan syndrome, its expression in mouse inner ear tissues was investigated by IHC [37,38]. Significant SLC4A11 expression was found in fibrocytes of the spiral ligament within the cochlea and in the vestibular labyrinth.

The localization and possible functions of SLC4A11 were investigated in mouse and rat salivary gland tissues, as high expression of SLC4A11 has been shown in this organ. Membrane localization was demonstrated [2,39]. Expression in the mouse submandibular gland was over eight times higher than that of other SLC4 family members, suggesting a specialized function. The SLC4A11 protein was localized to the basolateral membranes of acinar and ductal cells [39].

Overall, the cited studies show that *SLC4A11* mRNA is reliably detected only in select tissues. IHC staining confirms its presence in specific cell populations, predominantly localized to membranes. Therefore, *SLC4A11* expression should not be considered ubiquitous but rather specifically upregulated in diverse cell types across multiple tissues. The high expression in the corneal endothelium likely reflects the relative homogeneity of this tissue, which consists of a single cell layer lining the inner corneal surface.

### 2.4. SLC4A11 Expression in Pathologies

Several studies have compared *SLC4A11* gene expression in the corneal endothelium of FECD patients and healthy donors. The earliest study, using Serial Analysis of Gene Expression (SAGE), reported a significant decrease in *SLC4A11* expression in affected tissue [40]. In contrast, more recent investigations employing RNA sequencing and hybridization microarrays found increased *SLC4A11* expression in the corneal endothelium of FECD patients [41,42]. These conflicting results may arise from differences in disease stage among FECD patients and variations in the proportion of samples carrying expanded CTG18.1 trinucleotide repeats in the *TCF4* gene versus those without expansions. Distinct molecular mechanisms likely drive pathogenesis in these subgroups. For example, the study reporting elevated *SLC4A11* expression exclusively analyzed samples from patients with CTG18.1 expansion [41]. The absence of stratification by expansion status in other cohorts could mask differential expression patterns across genetic FECD subtypes. Transcriptomic analysis of corneal endothelium samples of FECD patients with and without CTG18.1 expansions showed no differential expression of *SLC4A11* between control and FECD groups [28].

Functional analyses of mutations associated with endothelial dystrophies have shown that many lead to reduced levels of SLC4A11 protein or its mature, glycosylated form [43].

Recent research has also revealed high SLC4A11 protein expression in various cancers, suggesting a role in tumor growth and progression. When *SLC4A11* was first described, the GeneBank database mentioned the discovery of *SLC4A11* cDNA fragments, expressed sequence tags (ESTs), corresponding to a sample of five pooled ovarian tumors and a sample of squamous cell carcinoma of the lung [1]. More recent reanalysis of hybridization microarray data demonstrated significant upregulation of *SLC4A11* in ovarian adenocarcinoma compared to normal ovarian epithelium [11]. Additionally, quantitative PCR studies have shown increased *SLC4A11* expression in hepatocellular carcinoma (HCC) [44] and gastric cancer [45]. Elevated *SLC4A11* expression has also been detected in colorectal cancer samples using various methods [46,47]. IHC analyses have confirmed SLC4A11 protein expression in ovarian, hepatocellular, gastric, and colorectal cancer tissues [11,44,45,46].

### 2.5. Structural Characteristics of the SLC4A11 Protein

The SLC4 family of transporters are integral N-glycosylated membrane proteins that are 850 to 1250 amino acids long and have a molecular mass of 90 to 200 kDa due to the presence of oligosaccharide fragments. All SLC4 polypeptides have three major domains: a cytosolic N-terminal domain (NTD) of 300−700 amino acids; an integral transmembrane domain of approximately 500 amino acids; and a small cytosolic C-terminal domain (CTD) of 40−130 amino acids [48]. In the case of SLC4A11, the intracellular location of the N- and CTD characteristic of the SLC4 family was confirmed. In addition, the location of 14 transmembrane sites of the polypeptide was determined and glycosylation sites within the extracellular loop EL3 between transmembrane sites 5 and 6 (545 and 553 amino acid residues of NP_114423.1) were predicted [49]. SLC4A11 has been shown to form oligomers within the plasma membrane, a characteristic of SLC4 family members. The importance of the cytoplasmic domain, especially the first 300 amino acid residues (NP_114423.1), for SLC4A11 dimerization has been established [50,51]. The cytoplasmic domain is also required for normal polypeptide chain stacking, which stabilizes the protein molecule; without it, SLC4A11 does not accumulate in cells [51].

## 3. Functional Features of the SLC4A11 Protein

### 3.1. Transport Function

#### 3.1.1. Initial Works

The *SLC4A11* gene was initially described, and its protein product characterized as a bicarbonate transporter based on its homology to known bicarbonate transporters [1]. Prior to 2004, no functional data on SLC4A11 were available [52]. Interest in SLC4A11’s function grew following the discovery of its high sequence similarity to AtBor1, a borate transporter in Arabidopsis thaliana. Studies showed that cells expressing *SLC4A11* exhibited increased membrane permeability to H^+^ or OH^−^ ions, resulting in enhanced sensitivity of intracellular pH (pH_i_) to changes in external pH (pH_e_). This permeability was independent of borate presence. Additionally, permeability to Na^+^ ions was demonstrated. Supporting the borate transport hypothesis, cells expressing SLC4A11 displayed greater pH changes in the presence of borate compared to controls [2]. However, this study used EIPA, an inhibitor intended to block sodium–hydrogen exchangers (Na^+^/H^+^ exchangers, NHE), which was later found to also inhibit SLC4A11 activity [16]. Subsequent research demonstrated that, unlike other members of the SLC4 family, SLC4A11 does not transport borate or bicarbonate ions [16,17,19].

#### 3.1.2. NH_3_- and pH-Regulated H^+^/OH^−^ Transport

To assess SLC4A11 function without interference from other ion exchangers, Ogando et al. used HEK293 cells, which have low endogenous levels of SLC4A11 and bicarbonate transporters, and PS120 cells, an NHE-deficient cell line [16]. They found that cells expressing *SLC4A11* exhibited increased cytoplasmic Na^+^ concentration and increased membrane permeability to Na^+^, H^+^ or OH^−^, and NH_4_^+^ ions. The data suggested that Na^+^ and H^+^ are transported in opposite directions (Na^+^ influx, H^+^ efflux) or that Na^+^ and OH^−^ are transported together into the cell (Table 2). However, subsequent studies reported no evidence for SLC4A11-mediated Na^+^ transport across the membrane [19,25], raising the possibility that the previously observed Na^+^ transport effects may be due to other membrane transporters.

Several studies, including those by Ogando et al. [16], Jalimarada et al. [17], and Zhang et al. [20], investigated changes in intracellular pH (pH_i_) following the addition and removal of ammonium chloride (NH_4_Cl) (Table 2). Using HEK293, PS120, and bovine corneal endothelial cells (BCECs), Ogando et al. [16] and Zhang et al. [20] observed a significantly faster decrease in pH_i_ in *SLC4A11*-expressing cells during NH_4_Cl incubation. Additionally, after NH_4_^+^ removal, these cells showed a more pronounced and rapid drop in pH_i_ compared to controls [16,20]. In Zhang et al. this effect was statistically significant [20]. Ogando et al. attributed this to increased membrane permeability of *SLC4A11*-expressing cells to NH_4_^+^ (entry of NH_4_^+^ into the cells leads to acidification of the medium) [16]. Zhang et al. concluded that SLC4A11 mediates the cotransport of NH_3_ and H^+^ (potentially bidirectionally) [20] (Table 2). After NH_4_Cl removal, pH_i_ recovery (alkalinization) was also faster in *SLC4A11*-expressing cells [20], which was consistent with the findings of Ogando et al. and Jalimarada et al. [16,17]. Ogando et al. suggested this was due to H^+^ efflux or OH^−^ influx via SLC4A11 [16].

Furthermore, Zhang et al. noted that the rate of ion flux associated with *SLC4A11* expression after NH_4_Cl removal was so low that non-ammonium-associated H^+^/OH^−^ transport by SLC4A11 could not be confirmed [20]. Zhang et al. also analyzed whole-cell currents at different pH_e_ and NH_4_Cl concentrations and failed to detect currents across the membrane when the pH_e_ was changed but no NH_4_Cl was added [20]. They concluded that there is no NH_3_-independent H^+^/OH^−^ transport across the membrane. In the presence of NH_4_Cl, membrane currents were influenced by the pH gradient. This study used isoform 1 of SLC4A11, which predominantly localizes to the cytoplasm [15], potentially limiting interpretation of the results. Later, the same group demonstrated NH_3_:H^+^ cotransport in transgenic mouse models and immortalized mouse corneal endothelial cell (MCEC) cultures (*Slc4a11*^+/+^ and *Slc4a11*^−/−^) (Table 2) [53].

Loganathan et al. provided evidence that the SLC4A11 protein transports NH_3_ across the membrane, but without additional transport of any ions (H^+^, Na^+^, NH_4_^+^) (Table 2) [19]. Using *Xenopus laevis* oocytes expressing *SLC4A11* and HEK293 cells transfected with *SLC4A11*, they observed enhanced NH_3_ transport in high-pH media. This was explained by a shift in the equilibrium between NH_3_ and NH_4_^+^ towards the formation of NH_3_ and H_2_O under these conditions [19]. In contrast, the work of another team, also using *Xenopus laevis* oocytes (mouse *SLC4A11* mRNA was used), demonstrated H^+^/OH^−^ transport independent of NH_3_/NH_4_^+^ presence, with no transport of other ions [18]. Myers et al. showed that the relationship between membrane potential and pH gradient followed the Goldman–Hodgkin–Katz equation for exclusive H^+^/OH^−^ transport (Table 2) [18]. Both pH_e_ and pH_i_ elevations increased membrane conductance. Changes in pH_e_ shifted the pK_i_ (the pH_i_ at which conductance of H^+^/OH^−^ ions is half-maximal), affecting H^+^/OH^−^ transport by SLC4A11. Mutations in the cytosolic domain of SLC4A11 caused significant shifts in pK_i_, critical for efficient transport [18,54,55]. It was concluded that the presence of NH_4_Cl affected membrane conductance indirectly, likely via membrane depolarization or an increase in pH_i_ [18].

Subsequent studies confirmed both SLC4A11-mediated H^+^/OH^−^ transport and the involvement of NH_3_/NH_4_^+^ in these processes. Kao et al. [21] confirmed the results of Myers et al. [18], who found that, in the absence of NH_4_Cl, the dependence of membrane potential on the pH gradient aligns with the Goldman–Hodgkin–Katz equation for exclusive H^+^/OH^−^ transport (Table 2). While most prior studies used the v2M1 isoform of SLC4A11, Kao et al. employed the v3 isoform [21]. In this work the effect of NH_3_/NH_4_^+^ was also assessed. Increasing pH_e_ to 8.0 and adding NH_4_Cl both induced greater pH_i_ changes in *SLC4A11*-expressing cells than controls; each factor independently elicited this effect [21]. It has been shown that in the presence of NH_3_ in the media, the relationship between the reverse membrane potential and the NH_3_ concentration fits the Goldman–Hodgkin–Katz equation for two competing membrane processes: H^+^/OH^−^ conduction and H^+^ cotransport with NH_3_ or as part of NH_4_^+^ ions (Table 2). However, passive NH_3_ diffusion through the membrane was not excluded [21].

In the most recent study examining H^+^/OH^−^ transport by SLC4A11 in the presence of NH_4_Cl, the coding RNA was produced on the matrix of a variant 2 transcript cloned into the pBSXG4 vector. It remains unclear whether the Kozak sequence at the first ATG codon was native or artificial; thus, the protein isoform could be either v2M36 or v2M1. This study showed that NH_3_/NH_4_^+^ act as allosteric activators of SLC4A11, affecting H^+^/OH^−^ conductance, decreasing pK_i_, and increasing H^+^ conductance under physiological conditions (Table 2) [57]. These results were obtained by the same group that previously demonstrated NH_3_/NH_4_^+^-independent SLC4A11-mediated H^+^/OH^−^ transport and concluded that NH_3_/NH_4_^+^ only indirectly affects the conductance of SLC4A11-containing membranes [18]. In addition, the influence of pH_i_ and pH_e_ on H^+^/OH^−^ transport was confirmed one more time in this work.

Multiple studies using various methods have shown that the cytosolic domain influences SLC4A11’s H^+^/OH^−^ transport capacity. Cryoelectron microscopy data indicate that mutations in this domain (e.g., NP_114423.1:p.(Arg125His)) alter protein conformation and H^+^/OH^−^ ion transport [54,55,56].

In conclusion, the investigation of the SLC4A11 transport function has yielded competing results concerning H^+^/OH^−^ transport and the role of NH_4_Cl in this process. It has been proposed that NH_4_Cl directly provides substrate (NH_3_/NH_4_^+^) for transport or cotransport with H^+^ or OH^−^, indirectly stimulates H^+^ conductance by altering cellular conditions (membrane potential and pH_i_), or does not affect SLC4A11 transport function at all. However, recent evidence has come to a limited consensus that NH_3_ and NH_4_^+^, as well as pH_i_ and pH_e_, affect H^+^/OH^−^ transport and likely act as allosteric activators of SLC4A11.

#### 3.1.3. Water Transport

SLC4A11 has been demonstrated to be implicated in the process of water transport. Vilas et al., using the *X. laevis* oocyte expression system, obtained evidence of SLC4A11-mediated transmembrane water movement that was similar in rate to some AQPs, uncoupled from solute-flux, inhibited by stilbene disulfonates (classical SLC4 inhibitors), and inactivated in one CHED2 mutant (NP_114423.1:p.(Arg125His)) [35]. Some *SLC4A11* variants have been shown to have an effect on water transport in the presence of an osmotic gradient (water flux assays) [58]. Later Kao et al. also provided results in favor of Na^+^-independent, electrically silent water translocation by SLC4A11 (Table 2) [15]. However, the underlying mechanism was not identified. No rigorous evidence for water movement through SLC4A11 has been obtained since. Perhaps water is not transported by SLC4A11 itself, but this protein indirectly affects water flux across the lipid bilayer.

#### 3.1.4. SLC4A11 and Its Function in Lactate-Mediated Corneal Endothelial Pump Activity

The cornea has a high level of glycolysis. Approximately 85% of glucose is metabolized to ATP [59,60]. Glycolysis is primarily performed by the surface epithelium and stromal keratocytes, which contain few mitochondria [61]. During eye closure or contact lens wear, corneal oxygenation is reduced to one-third of atmospheric concentration, creating hypoxic conditions [62,63]. A shift to anaerobic metabolism with lactate accumulation has been shown under hypoxic conditions [64,65]. Lactate removal across the ocular surface is difficult due to the multilayered epithelium [66]. It has been shown that lactate, rather than HCO_3_^−^, is the key ion that drives the endothelial fluid pump [67,68]. Active transport of lactate by endothelial cells towards the anterior chamber allows the pumping of water (moving together with dissolved substance) from the corneal stroma, thus preventing its edema [69]. An intact osmotic barrier is also a prerequisite for effective lactate-mediated endothelial fluid pump function [68,69].

Lactate transport across the corneal endothelial membrane is mediated by the transporters MCT1, MCT2, and MCT4. MCT1 and MCT4 are part of the basolateral membrane complex with the chaperone CD147. These transporters function as part of the basolateral membrane to provide lactate transport into the cell coupled with H^+^ [67,70,71]. It has also been shown that MCT2 in bovine and rabbit endothelial cells is located within the apical membrane [67,72]. In contrast, it effluxes lactate and H^+^ into the anterior chamber.

In conventional *SLC4A11* KO mouse models, MCT4 gene expression has been shown to be reduced [73]. It has been hypothesized that this leads to lactate accumulation. Indeed, increased levels of lactate have been found in the stroma of both conventional and inducible *SLC4A11* KO mouse models [73,74]. In the inducible *SLC4A11* KO mouse model, the expression levels of the lactate transporters MCT2 and MCT4 proteins were shown to be decreased in mouse corneal endothelial cells [75]. Interestingly, the mitochondrial antioxidant visomitin or the Src kinase inhibitor eCF506 rescued MCT2 and MCT4 expression in an inducible *SLC4A11* KO mouse model. Authors proposed that this effect may be mediated by decreasing ROS production. However, the mechanism is unknown [75]. In the *SLC4A11* KO mice model, AAV9-HA-Slc4a11 injection caused a 30% reduction in corneal lactate and significant increase in MCT4 expression in young animals [76].

In 2016, Nehrke and Parker proposed the initial mechanism of SLC4A11 involvement in lactate-mediated corneal endothelial pump activity [18,77]. In 2022, Bonanno et al. further detailed this by linking SLC4A11 activity to MCT4, as they are both located on the basolateral membrane [12]. SLC4A11 promotes the corneal endothelial pump through an H^+^ buffering mechanism that enhances lactate transport. Lactate influx from the stroma to the endothelial cell through MCT1/MCT4 and lactate efflux through MCT2 into the anterior chamber may induce a local H^+^ deficit (pH_i_ change). SLC4A11 as a pH sensor activates H^+^ influx into the corneal endothelium. This in turn increases the availability of H^+^ ions for MCT transporters. Consistent with the results in mouse models, it was proposed that SLC4A11 specifically facilitates MCT4 lactate:H^+^ efflux from corneal endothelial cells into the lateral space.

SLC4A11 has emerged as an important regulator of the lactate-mediated corneal endothelial pump. Its activity supports H^+^ buffering, which enhances lactate transport by MCTs, particularly MCT4 on the basolateral membrane. The exact molecular interplay between SLC4A11 and MCT4 remains to be fully elucidated, as does the relationship between MCTs and Src kinase inhibitors.

### 3.2. SLC4A11 and Oxidative Stress

The role of SLC4A11 in oxidative stress and mitochondrial dysfunction has been of increasing interest to several research groups because of its potential contribution to the pathogenesis of corneal endothelial dystrophies. This chapter reviews the current evidence on the role of SLC4A11 in protecting corneal endothelial cells from oxidative damage, its interaction with the NRF2 antioxidant pathway, and the downstream effects on mitochondrial health, autophagy, and endoplasmic reticulum (ER) function.

Multiple studies have demonstrated markers of oxidative stress in FECD corneas. For example, when gene expression was examined in the corneas of FECD patients by the SAGE method, decreased expression of the antioxidant enzyme glutathione-S-transferase P gene was revealed [40]. IHC staining showed nitrotyrosine accumulation characteristic of oxidative stress [78], excessive glycation products, and advanced glycosylation end-products (AGE) [79]. Mitochondrial DNA damage and oxidative stress-induced redox imbalance have also been reported in the corneal endothelium of patients with FECD [80,81]. However, the involvement of SLC4A11 dysfunction in the pathogenesis of FECD should be treated with caution as its causal role in FECD is questionable (see Section 4.2. for details).

Sanhita Roy’s work suggests that SLC4A11 may play a role in counteracting oxidative stress [82]. Mutations in *SLC4A11* associated with CHED result in increased generation of reactive oxygen species (ROS) and mitochondrial dysfunction. In these cases, the expression of key antioxidant genes (*NRF2*, *HO-1*, and *NQO1*) is significantly reduced, and a higher rate of apoptosis is observed under oxidative stress conditions. 

Further studies have confirmed that *SLC4A11* depletion is associated with increased ROS production, exacerbating cellular dysfunction and apoptosis [83]. Using siRNA to knock down *SLC4A11* in primary human corneal endothelial cell culture, researchers found that reduced *SLC4A11* expression increases cellular sensitivity to oxidative stress, leading to greater damage and apoptosis [83]. This effect appears to be due to SLC4A11’s role in regulating NRF2, a key transcription factor that controls the expression of several genes responsible for suppressing oxidative stress. Under oxidative stress, NRF2 is activated and translocates to the nucleus, where it initiates transcription of antioxidant genes such as *HO-1* and *NQO1*. Decreased *SLC4A11* expression leads to lower NRF2 and downstream antioxidant gene expression, impairing the cell’s ability to counteract oxidative stress and resulting in increased ROS and cytotoxicity. The precise mechanism by which SLC4A11 regulates antioxidant signaling remains an area of future research and is not yet fully understood.

SLC4A11 is also crucial for maintaining mitochondrial integrity. Loss of *SLC4A11* has been shown to cause mitochondrial membrane depolarization, an early event in apoptosis [22]. Further release of cytochrome c from mitochondria into cytoplasm enhances apoptotic signaling. Thus, the protective role of SLC4A11 is demonstrated by its ability to maintain normal mitochondrial function and reduce ROS generation [83].

Given the involvement of SLC4A11 in oxidative stress, the observed decrease in NRF2 expression in the corneal endothelium of CHED patients may be explained by oxidative stress resulting from impaired or complete loss of SLC4A11 function caused by mutations in the corresponding gene [83]. In a later study, the positive feedback of Nrf2 and *SLC4A11* expression was demonstrated. It was shown that Nrf2 affects *SLC4A11* expression at the level of transcription regulation by directly binding to specific sites, antioxidant response elements (ARE), in the promoter region of the *SLC4A11* gene [30]. The study identified three ARE sequences located upstream of the *SLC4A11* transcription start site. When the Nrf2 signaling pathway is activated by tert-butylhydroquinone (tBHQ), Nrf2 binds to these ARE sites, resulting in increased transcriptional activity of *SLC4A11*. Nrf2 binding to the promoter enhances *SLC4A11* expression, as evidenced by increased mRNA levels and elevated SLC4A11 protein levels in both HeLa cells and human corneal endothelial cells. In contrast, depletion of Nrf2 using a specific siRNA significantly reduced *SLC4A11* expression, further supporting the role of Nrf2 as a critical regulator of *SLC4A11* transcription under conditions of oxidative stress [30].

The corneal endothelium has a high density of mitochondria to provide energy for the active transport necessary for corneal hydration. Endothelial cells utilize glucose and glutamine as a source of energy, and a deficiency of glutamine leads to a decrease in ATP levels and a reduction in the transport function of the endothelial cells. The use of radiolabeled 12C-glutamine showed that about half of the carbon in the composition of tricarboxylic acid cycle intermediates was previously incorporated into glutamine molecules [84]. As a result of active glutamine catabolism in endothelial mitochondria, the NH_3_ concentration should increase, the load on the electron transport chain should also increase, and the proton gradient in the intermembrane space should accumulate. At the same time, SLC4A11 has been reported to be able to actively transport H^+^ in the presence of NH_3_. In this context, Ogando et al. suggested that SLC4A11 may act as a mitochondrial uncoupler activated by increased glutamine catabolism [22].

SLC4A11 was identified as part of the inner mitochondrial membranes of human (IF staining and Western blot analysis) and mouse (Western blot analysis) corneal endothelial cell cultures [22]. It was shown that exposure of wild-type (WT) corneal endothelial cells to ammonium acetate significantly depolarized the mitochondrial membrane potential, whereas no such response was detected in *SLC4A11* knockout (KO) cells. This suggests that SLC4A11 plays a role in the regulation of mitochondrial membrane potential. Glutaminolysis fuels the tricarboxylic acid cycle by producing reducing equivalents that accelerate the electron transfer chain. This has been shown to increase superoxide (O_2_^−^) radical formation, as has the direct action of NH_3_ on complexes I and III. The rapid conversion of O_2_^−^ to hydrogen peroxide is a hallmark of this process. It has been hypothesized that NH_3_ either directly activates SLC4A11 or diffuses across the inner membrane and cotransports H^+^ into the matrix, while the uncoupling action (movement of excess H^+^ from the intermembrane space into the mitochondrial matrix) of SLC4A11 prevents extreme mitochondrial membrane hyperpolarization, excess O_2_^−^ generation, mitochondrial damage, and apoptosis [12,22,85].

When *SLC4A11* expression was suppressed, a decrease in glutamine catabolism was shown in corneal endothelial cells [84]. Under SLC4A11-deficient conditions, glutamine (in vivo and in vitro) induced ER stress, which was reversible when mitochondrial ROS levels are reduced [86]. In cultured *SLC4A11* KO endothelial cells, impaired autophagy flux was detected. As in WT cells, the level of phosphorylated mTOR protein was decreased compared to non phosphorylated mTOR, and the level of the LC3bII subunit of the light chain of microtubule-associated protein 1A/1B was increased compared to LC3bI, which is characteristic for the initial stages of autophagy. However, there was no decrease in the level of P62, which is characteristic of active autophagy—the level of this protein in KO cells was significantly higher than in WT cells. At the same time, the level of P62 in WT cells increased to about the same level when the cells were treated with bafilomycin A1 (BafA1), which blocks the pumping of H^+^ into lysosomes and prevents the digestion of P62 and other proteins in them.

Similar phenomena were observed in the *SLC4A11* KO MCECs. A decrease in the expression of lysosomal proteins, vacuolar ATPase and cathepsin B (lysosomal hydrolase), was also observed in MCECs. In addition, the level of TFEB, a regulator of the transcription of genes related to autophagy and lysosomal function, was significantly reduced in the nuclear fraction of the MCECs. An overall decrease in the level of TFEB was demonstrated in the *SLC4A11* KO MCECs.

In the absence of glutamine, but with treatment with the antioxidant mitoQ, the KO cells showed normalization of the progression of autophagy, including end steps associated with lysosomal digestion. There was also a decrease in corneal thickness and ROS levels, as well as a normalization of markers of normal lysosomal function in mice after intraperitoneal injections of mitoQ. Thus, it was considered that oxidative stress led to impairment of the autophagy process [86].

The source of lysosomes in cells is the ER. Using the MCEC cell line *Slc4a11*^−^/^−^, it has been shown that ROS-induced ER stress leads to lysosomal dysfunction and impaired autophagy. One manifestation of ER stress is impaired calcium (Ca^2+^) accumulation in the ER. It manifested by reduced Ca^2+^ release from the ER in *Slc4a11*^−^/^−^ cells upon glutamine stimulation and was associated with mitochondrial production of ROS. The increased ROS levels, impaired autophagy, and ER stress observed in the SLC4A11 deficiency model were similar to the pathophysiological changes in CHED, suggesting a potential role for ROS-mediated ER stress in the pathogenesis of CHED [87].

It has been hypothesized that the transport of SLC4A11 protein into mitochondria occurs at the protein maturation stage. SLC4A11 does not have a mitochondrial localization signal. SLC4A11 precursors synthesized on ER-associated ribosomes interact with the cytosolic chaperones HSP90 and/or HSC70 to direct the protein to the mitochondria through specific binding to the TOM70 receptor on the outer mitochondrial membrane. After translocation across the outer mitochondrial membrane by the TOM complex, SLC4A11 may be transported across the intermembrane space to the inner mitochondrial membrane by the TIM9-TIM10 complex. The mechanism of SLC4A11 release to the inner mitochondrial membrane remains unclear [88].

Recently, new data have reported that in corneal endothelial cells, SLC4A11 is weakly co-localized with the mitochondrial marker COX4 and is predominantly located at the plasma membrane. In addition, SLC4A11 was co-localized with COX-4 mainly as part of punctate structures that may be formed during mitochondrial autophagy (mitophagy) [36].

Interestingly, histological examination of the corneal endothelium did not show reduced cell density in all cases of CHED [89]. The decrease in corneal endothelial cell density in *SLC4A11* KO mice was either insignificant [37] or developed later compared to the corneal edema [74,90]. Perhaps the primary result of SLC4A11 dysfunction is corneal edema, but not oxidative-stress-mediated cell death. However, oxidative stress products that accumulate as a result of *SLC4A11* KO may alter regulatory pathways in endothelial cells, affecting endothelial barrier properties as well as processes related to water efflux from the corneal stroma (ATP production, lactate transport) [74,75].

In summary, SLC4A11 serves as a regulator of oxidative stress, mitochondrial function, and autophagy in corneal endothelial cells. SLC4A11 function decreases the levels of ROS, which are produced in large amounts during glutamine catabolism. In this case, under conditions of oxidative stress, SLC4A11 and Nrf2 regulate each other’s expression in a positive feedback loop. *SLC4A11* expression in cells that actively use glutamine as an energy source has been shown to be necessary for normal ER function and autophagy. However, recent evidence showing weak co-localization of SLC4A11 with mitochondrial markers and its presence in autophagosome-like structures highlights the need for further research into its co-localization with cellular compartments involved in autophagy and mitophagy [36]. The primary role of the oxidative stress pathway in the pathogenesis of CHED caused by mutations in *SLC4A11* is disputable.

### 3.3. Role in Cell Adhesion and Effects on Cell Proliferation and Viability

Given the primary impairment of corneal endothelial barrier function in the development of CHED, the association of SLC4A11 protein with cell adhesion seems particularly important. Loss of corneal endothelial cells is observed in CHED, which may be associated with impaired cell adhesion mechanisms [74,75,90]. This is also supported by the predominantly basolateral localization of SLC4A11 in corneal endothelial cells [35,36].

Several studies have shown that isoform 2 of the SLC4A11 protein can function as cell adhesion molecules in the corneal endothelium. Malhotra et al. investigated the role of SLC4A11 in the adhesion of corneal endothelial cells to Descemet’s membrane [23,24]. The interaction of SLC4A11 with Descemet’s membrane through the third extracellular loop (EL3) was shown to be mediated by COL8A2 and COL8A1 proteins [23,24]. Thus, *SLC4A11* mutations may prevent the normal attachment of endothelial cells to Descemet’s membrane, leading to corneal edema and endothelial cell loss. However, to assess the contribution of SLC4A11 protein to the attachment of endothelial cells to Descemet’s membrane and the maintenance of the integrity of the endothelial monolayer, further studies are needed, especially considering the presence of specialized cell adhesion molecules [91].

Abnormal cell morphology and loss of adhesion were observed in the corneal endothelium of *SLC4A11* KO mice [74,75,90]. Immunostaining was used to show disruption of the structure of both the outer and inner parts of tight junctions in the corneal endothelium when SLC4A11 protein was knocked out [74,75]. To maintain the normal structure and function of intercellular junctions, the normal formation of their intracellular part is necessary [92,93,94]. Both the tight junction membrane protein ZO-1 and intracellular F-actin have a much less ordered arrangement in the absence of functional SLC4A11 [74,75].

The likely association of SLC4A11 with normal cytoskeleton formation in the corneal endothelium is also suggested by the abnormal morphology of endothelial cells found in histologic examination of corneal endothelium from patients with CHED, and the presence of multinucleated cells [89].

### 3.4. Association of SLC4A11 with Regulation of Cellular Processes

Two studies have examined the effects of *SLC4A11* KO in mice (by deleting exons 9–13) using transcriptome analysis [73,95]. Alvarez et al. [95] compared the total corneal transcriptome of 17-week-old *SLC4A11* KO and wild-type (WT) mice, while Ogando et al. [73] focused specifically on the corneal endothelial transcriptome of 12-week-old mice. Both studies found an enrichment of differentially expressed genes related to cell adhesion and the cytoskeleton. Notably, the study by Alvarez et al. identified a large number of differentially expressed genes responsible for cell fate determination and development [95]. These include transcriptional regulators (*fhl3*, *glis3*, *tshz3*, *sox4*, *dlx5*, *tox*, *zbtb16*), components of developmental signaling pathways (*sostdc1*, *srgap1*, *vtcn1*, *dkk1*, *dlk2*, *areg*, *gpr161*, *smo*, *tll1*, *tnfrsf11b*, *smim31*, *shisa2*, *cntfr*, *rasl11b*), and cell fate specificity factors (*nol4*, *tnfaip2*, *kprp*, *ccdc88c*, *lrrc4*, *clec2g*). The authors suggested that SLC4A11 may have a regulatory role in these processes. Ogando et al. also reported significant changes across multiple pathways, attributing these alterations to molecular damage caused by ROS generated during oxidative stress [73].

Both studies showed similar enrichment in genes related to cell signaling. Ogando et al. [73] highlighted genes involved in G protein-coupled receptor signaling, cAMP-mediated signaling, and calcium signaling, consistent with Alvarez et al.’s [95] findings of enrichment in calcium ion binding, multicellular organism development, and adenylate cyclase-activating G protein-coupled receptor pathways. Further investigation by Ogando et al. linked changes in cellular processes in corneal endothelium of *SLC4A11* KO mice to c-SRC kinase activation under oxidative stress conditions [75].

In the work of Ogando et al. the most significant terms in the enrichment analysis were cholesterol synthesis and glycolysis pathways [73]. Ogando et al. found that cholesterol synthesis genes were significantly upregulated in *SLC4A11* KO corneal endothelium, whereas glycolysis-related genes were downregulated. Less active cholesterol synthesis in cells expressing *SLC4A11* may explain the observed [14,35] increase in cell membrane permeability to water. In addition, glycolysis and lactate transport appear to play important roles in corneal water flux [67,76]. Another transcriptomic study (on cultured corneal endothelial cells) also showed decreased glycolytic enzyme expression in the absence of SLC4A11 [96].

A recent publication [36] validated data from the literature regarding the various effects of *SLC4A11* loss or mutation. Decreased cell viability, increased proliferation and migration, impaired barrier function, and abolition of NH_3_-induced membrane conductance were shown. Interestingly, low proliferation capacity, high adhesiveness, and strong barrier properties are hallmarks of normal corneal endothelium, and together with cell survival are regulated during cell fate determination. Thus, the involvement of SLC4A11 in cell fate regulation seems quite likely. The same work showed a weak co-localization of SLC4A11 with inner mitochondrial membrane molecules, which does not support the effect of SLC4A11 on a variety of biological processes by alleviating oxidative stress [36].

How might SLC4A11 regulate corneal endothelial cell fate? We will make assumptions based on the preferential membrane localization of SLC4A11. Moreover, localization on the basement membrane side and interaction with Descemet’s membrane collagens have been shown. Cell adhesion molecules (e.g., integrins or cadherins), when interacting with the intercellular matrix or other cells, are able to determine cell fate. It is possible that SLC4A11 contributes to the proper function of these proteins within the corneal endothelial plasma membrane. In addition, SLC4A11 maturation occurs in the ER (it is a transmembrane protein, and ER accumulation of mutant forms has been observed [3]), and SLC4A11 has also been shown to be associated with normal lysosome formation, normal ER function, and the ability to accumulate Ca^2+^ [87]. It is possible that SLC4A11 enables the normal functioning of EPR-related processes and thus normal cell fate determination. MFSD1 (SLC72A1) is an example of the influence of a solute carrier protein on the regulation of cellular processes. By decreasing the intensity of lysosomal degradation of inactive integrin, MFSD1 leads to an increase in its concentration at the membrane [97]. As a result, the ratio of active and inactive forms in the membrane is reduced, leading to increased cell adhesion.

Cholesterol synthesis begins with acetyl-coenzyme A, derived from mitochondria and transported to the cytosol. The regulation of the activity of this process (including the expression of the necessary enzymes) is related to the function of the ER and the transport of SREBP from the ER to the Golgi apparatus [98]. Transcriptional upregulation of cholesterol synthesis is observed in lysosomal defect models, whereas mitochondrial defects lead to its downregulation [99]. Thus, the dysregulation seen in SLC4A11-deficient cells likely arises primarily from ER stress.

The involvement of SLC4A11 in the regulation of intracellular processes and the establishment of the endothelial phenotype is currently only an assumption, which should be confirmed experimentally in the future.

## 4. SLC4A11 Functions in Pathologies

### 4.1. SLC4A11 Variants in CHED

Two primary pathologies associated with the pathogenic variants in *SLC4A11* protein are CHED and Harboyan syndrome. CHED is an autosomal recessive disorder that typically manifests at birth and is more common in populations with a high rate of consanguineous marriages. In Harboyan syndrome, corneal dystrophy is accompanied by progressive sensorineural hearing loss [100].

Clinically, CHED presents as bilateral, symmetrical, noninflammatory corneal edema. This edema appears as a diffuse, blue-gray, ground-glass-like opacification in otherwise healthy, full-term newborns [29]. The most common additional clinical sign is nystagmus. Edema of the corneal epithelium progresses from fine swelling without bullae to the formation of subepithelial bullae [101]. Histopathological examination of corneal buttons reveals ruptures in Bowman’s membrane in cases of advanced edema. Transmission electron microscopy reveals a two- to threefold thickening of the stroma with severely disorganized lamellae. Descemet’s membrane is uniformly thickened. The density of corneal endothelial cells is usually reduced in the central cornea, while the peripheral cornea may retain relatively normal cell density [101]. In Harboyan syndrome, corneal opacification resembles that seen in CHED. Hearing loss typically develops postlingually, although a case of prelingual hearing loss has been reported [102]. In some patients initially diagnosed with CHED, subsequent identification of hearing loss led to a revised diagnosis of Harboyan syndrome [3,6,103].

In 1999, a genetic linkage study identified a locus on chromosome 20p13 associated with autosomal recessive CHED in a large consanguineous Irish family [104]. A critical region of homozygosity spanning 8 cM was identified, which later overlapped with the locus (CDPD1) associated with Harboyan syndrome in a Moroccan family [105]. In 2006, mutations in the *SLC4A11* gene within this region were identified as the cause of autosomal recessive CHED [3]. Functional experiments revealed that the identified variants either disrupted membrane targeting or triggered nonsense-mediated decay. The following year, an article describing homozygous or compound heterozygous *SLC4A11* mutations in patients with Harboyan syndrome was published [5]. To date, more than 100 variants of the *SLC4A11* gene have been reported, with approximately 80 percent of patients having homozygous or compound heterozygous variants in this gene [3,4,5,6,7,29,102,103,106,107,108,109,110,111,112,113,114,115,116,117,118,119,120,121,122,123,124,125,126,127,128,129,130,131,132,133,134,135,136,137,138,139,140,141].

We summarized the data on the reported CHED and Harboyan syndrome variants in Appendix A and Figure 2. The majority of the studies cited above reported *SLC4A11* variants using transcript variant 2 (NM_032034) and its corresponding protein isoform v2 (NP_114423). To maintain consistency with these reports, we initially annotated coding sequence and amino acid changes relative to this transcript. However, transcript variant 3 (NM_001174089.2) and isoform v3 (NP_001167560.1) have been designated as the MANE (Matched Annotation from NCBI and EMBL-EBI) Select reference, reflecting their status as the biologically representative isoforms. Recent studies have identified SLC4A11-v3 and SLC4A11-v2M36 as two major protein isoforms expressed in the corneal endothelium [14,25,28]. Notably, SLC4A11-v3 fully contains the polypeptide sequence of the second major endothelial isoform, SLC4A11-v2M36. Therefore, in Appendix A and Figure 2 we utilized NP_001167560.1 (isoform v3) to indicate the locations of CHED-associated mutations in SLC4A11. Also, we provided parallel annotations for all reported variants in transcript variants 2 (NM_032034) and 3 (NM_001174089) within Appendix A.

The majority of CHED patients reported in the literature are from South Asia (particularly India and Pakistan) and the Middle East (see Appendix A). In contrast, cases of CHED associated with *SLC4A11* variants have been rarely reported in Central America, Australia, and New Zealand. Homozygous or compound heterozygous *SLC4A11* variants were detected in patients with Harboyan syndrome. No obvious pattern of mutations distinguishing CHED from Harboyan syndrome was found [100]. In some cases, the same variant was reported for both conditions (see Appendix A). However, genetic analyses in some CHED cases revealed no candidate *SLC4A11* variants or only a single heterozygous mutation [29,127,133,134,136,143]. This incomplete detection could stem from limitations in conventional sequencing methods, such as Sanger sequencing, which cannot identify large genomic deletions. For example, a large deletion encompassing *SLC4A11* was identified in a Thai family with CHED using array comparative genomic hybridization [130].

The most prevalent variants in CHED are NP_001167560.1:p.(Arg853Cys), NP_001167560.1:p.(Arg142Glnfs*4), NP_001167560.1:p.(Glu659Ala), NP_001167560.1:p.(Val808Met), and NP_001167560.1:p.(Arg739Gln) (see Figure 2). Many of these variants are located in the transmembrane regions or between transmembrane regions and other domains. Of interest, variants in extracellular loops are rare.

### 4.2. Limited Evidence for a Causal Role of SLC4A11 Variants in FECD

Some studies have suggested that rare heterozygous *SLC4A11* variants may be implicated not only in CHED but also in FECD [8,9,58]. To confirm the presence of pathogenic variants, our systematic review collected all reports of *SLC4A11* variants detected in patients diagnosed with FECD [10]. In Appendix A and Figure 2 we present some of the *SLC4A11* variants from the systematic review [8,9,58,144]. We excluded variants in introns, synonymous variants, and variants found in the control groups of the studies. We also excluded studies without detected *SLC4A11* variants, studies describing variants with non-significant associations, and purely functional studies. Some FECD variants are located next to CHED/Harboyan syndrome variants, but they do not match completely except at one position: NP_001167560.1:p.224. However, the variants themselves differ: a variant leading to stop codon formation was detected in CHED, while a missense mutation was detected in FECD (Appendix A). The most prevalent variant described in FECD patients was reported in the Minear et al. study (NP_001167560.1:p.(Asn134Ser)) (Figure 2) [144]. Since there was no control group in this study, we formally included it in the table. However, according to GnomAD v4.1.0, the allele frequency of this variant in African Americans is 0.04266, and it was classified as benign by Varsome in our systematic review [10,145]. 

Seven variants identified in the *SLC4A11* gene in patients with FECD were classified as pathogenic [10]; however, there is insufficient evidence to confirm their causal role in FECD. Notably, segregation of *SLC4A11* variant carriage with the FECD phenotype was convincingly demonstrated in only a single family [9]. Another familial case reported one proband’s relative exhibiting the FECD phenotype; however, the genotype was not established in this instance [8]. In both reports, the CTG18.1 trinucleotide repeat expansion status was not determined, because the association of CTG18.1 repeats with FECD was unknown at the time of these investigations [146]. Therefore, a more common etiology in these cases cannot be excluded. A recent study by Liu et al. examined *SLC4A11* variants in a cohort of FECD patients who did not have expansions of the CTG18.1 trinucleotide repeats [147]. Unfortunately, variants in the *SLC4A11* gene were detected in sporadic, non-familial cases, thereby precluding segregation analysis [147].

A significant argument against a causal role for *SLC4A11* variants in FECD arises from clinical observations in parents of CHED patients. CHED results from homozygous or compound heterozygous variants, typically inherited from heterozygous carrier parents. If heterozygous *SLC4A11* variants contributed to FECD, one might expect clinical signs of FECD in these parents. However, investigations into FECD manifestations among CHED parents are limited. Three studies have addressed this question: Kim et al. examined one family, Chaurasia et al. studied eight families, and Siddiqui et al. evaluated three families [6,110,118]. Across all studies, parents available for clinical examination were asymptomatic. At least one parent in the families reported by Kim et al. and Chaurasia et al. exhibited cornea guttata [110,118]. In Siddiqui et al.’s work, two families had a parent with cornea guttae, whereas in the third family only one parent was available and showed no guttae [6]. Making a definite diagnosis was even more complicated given the age of the examined parents. The mean age of parents in Chaurasia et al.’s study [110] was 32.5 years, younger than the typical age of late-onset FECD manifestation, while the mean age in Siddiqui et al.’s study [6] was 41 years. Furthermore, critical diagnostic parameters such as the Krachmer grading score or endothelial cell density were unavailable for the oldest examined relative (a 62-year-old mother) in Kim et al.’s study [118], limiting conclusive assessment. Given these limitations, it remains challenging to definitively assess segregation of *SLC4A11* variants with the FECD phenotype in these cases.

Clinical interpretation of cornea guttata without associated symptoms or family history is a debatable topic [148,149,150]. Evidence suggests that a few scattered, non-confluent guttae (Krachmer stages 1–2) may represent an age-related phenomenon. Isolated guttae were detected in a substantial proportion of middle-aged individuals [151,152,153]. Consequently, the presence of guttae alone in CHED parents does not provide strong evidence for FECD. The authors of these studies have also expressed caution in their conclusions [6,110,118]. Taken together, the limited familial segregation data, the lack of exclusion of the CTG18.1 expansion in reported familial cases, and the absence of clear FECD clinical signs in heterozygous *SLC4A11* variant carriers, cast doubt on a direct causal relationship between *SLC4A11* variants and FECD.

### 4.3. Mouse Models of CHED

Four variants of *SLC4A11* KO mouse models have been described.

Gene Trap KO Model [37]: In the earliest model, mice deficient for Slc4a11 were generated using a retroviral gene trap vector integrated upstream of exon 2. The vector included a splice acceptor, neomycin resistance gene, and polyadenylation signal, with stop codons in all reading frames to terminate translation. Despite this disruption, corneal endothelial cell density and morphology remained normal in these mice, unlike the severe phenotypes seen in humans with *SLC4A11* mutations. However, abnormal epithelial and stromal morphology were observed.

β-Galactosidase Knock-In Model [38]: Another mice model was generated by inserting a β-galactosidase coding sequence in-frame into the 10th exon of *SLC4A11*. This mutation leads to the truncation of the SLC4A11 protein before the first predicted transmembrane domain resulting in a cytoplasmic localization of the β-galactosidase fusion protein in vivo. KO mice exhibited thickened corneal layers (endothelium, Descemet’s membrane, stroma, and epithelium), endothelial vacuolation, and disrupted epithelial cell shape. In this model, more severe corneal morphologic changes than in the gene trap KO model may be due to incomplete disruption of Slc4a11 by the gene trap mutation or to differences in the pathologic stages analyzed.

Cre-Lox Model [90]: Many studies have used these *SLC4A11* KO mice [35,73,95]. Exons 9–13 were flanked by loxP sites, and Cre-mediated deletion induced a frameshift with a premature stop codon in exon 16. The truncated mRNA underwent nonsense-mediated decay. The cornea from the KO mice was significantly thicker than that from the WT mice of the same age at all time points (from 10 to 40 weeks), and these differences increased with age. Increased cell size and decreased endothelial cell density were noted in KO mice at 40 weeks. Electron microscopy revealed age-dependent endothelial swelling, hexagonal morphology distortion, and eventual membrane destruction. In the KO mouse, mild endothelial cell swelling was observed at 10 weeks, although hexagonal morphology and clear cell borders were maintained.

Inducible Cre-Lox KO Model [74]: To study early events following loss of Slc4a11 activity, an inducible *SLC4A11* KO model (similar to the previous one) was generated. *SLC4A11* exons 9-13 were excised by the estrogen receptor–Cre recombinase fusion protein activated by tamoxifen [74]. Corneal thickness increased gradually and was 50% greater than WT after 8 weeks, with significantly altered endothelial morphology, increased nitrotyrosine staining, significantly higher stromal lactate, decreased expression of lactate transporters and Na-K ATPase activity, higher ATP, altered expression of tight and adherens junctions, and increased fluorescein permeability. A significant increase in corneal thickness was observed at the first time point 14 days after tamoxifen treatment. There was no significant difference in endothelial cell density, but there were significant differences in cell morphology.

Clinical features of Harboyan syndrome caused by loss-of-function mutations in *SLC4A11* in humans correlate quite well with those observed in *SLC4A11* KO animal models [37,102]. In humans with Harboyan syndrome, corneal changes are observed at birth, while sensorineural hearing loss is defined postlingually with typical onset in the second decade of life [137]. In a Gene Trap KO mouse model, loss of the SLC4A11 protein led to inner ear pathology. The only microscopic change in the mouse ear was a shrinkage of the cochlear membranous labyrinth [37]. This suggests a deficiency of endolymph due to an impaired transport function of either the fibrocytes or the stria that support their formation. However, a more recent study did not find any structural abnormality in the membranous labyrinth, but did find disturbances in the morphology of the fibrocytes, possibly due to osmolar stress [38].

As mentioned above, *SLC4A11* is reliably expressed in the kidney in both humans and mice. However, mutations in *SLC4A11* are unlikely to cause renal pathology in humans. Only one case of unilateral renal agenesis has been reported in a patient with Harboyan syndrome [105]. Other genetic causes of renal agenesis were not investigated in this case, so an association between *SLC4A11* mutation and renal agenesis is inconclusive. Hydronephrosis was reported in one patient [102]. But the other studies did not report kidney disease in patients with CHED or Harboyan syndrome, even in a targeted study [123,137].

In various models, the *SLC4A11* KO mice exhibited significant corneal edema, which appears to be influenced by impaired lactate efflux from the corneal stroma. Abnormal cell morphology, cell adhesion defects, decreased barrier function, and increased fluorescein permeability were also reported. In these models, the decrease in corneal endothelial cell density in *SLC4A11* KO mice was either insignificant [37] or developed later compared to the corneal edema [74,90]. The authors suggest that mitochondrial oxidative stress is associated with loss of *SLC4A11* [76]. Possibly, the primary consequence of SLC4A11 dysfunction is corneal edema, but not oxidative-stress-mediated cell death. However, oxidative stress products that accumulate as a result of *SLC4A11* KO may alter regulatory pathways in endothelial cells, affecting endothelial barrier properties as well as processes related to water efflux from the corneal stroma (ATP production, lactate transport) [74,75].

### 4.4. Functional Impacts of SLC4A11 Gene Mutations

Studies investigating mutations in the *SLC4A11* gene have revealed the following main mechanisms underlying protein dysfunction. A substantial subset of these mutations directly impairs the transport activity of the SLC4A11 protein without compromising its localization to the plasma membrane. Conversely, another group of mutations disrupts proper protein folding, thereby hindering its trafficking to the plasma membrane and reducing the availability of functional transporter at the cell surface [8,10,43,58,96,154]. The specific location of mutations within the SLC4A11 protein critically influences their effects on protein function and stability, thereby providing insights into the distinct roles of each domain. Mutations within the transmembrane domain can directly alter ion channel structure or charge properties, resulting in diminished ion permeability and loss of function. Similarly, mutations in the channel gate domain may disrupt the regulation of channel gating, adversely affecting ion flux and cellular homeostasis. Mutations in the NTD often impair regulatory interactions and the protein’s responsiveness to cellular signals or lipid binding, potentially leading to aberrant transporter activity. Notably, the cytosolic domain has been implicated in oxidative stress responses mediated by the Nrf2 pathway and in mitochondrial trafficking [56]. Finally, the cytoplasmic domain of SLC4A11 plays a crucial role in maintaining membrane domain stability and is essential for water flux activity through SLC4A11, possibly by facilitating substrate access to the cytosol [51]. Collectively, these findings underscore the importance of the cytosolic domain in sustaining SLC4A11 functional activity.

### 4.5. Role of SLC4A11 in Cancer

Increased expression of *SLC4A11* has been demonstrated in several cancer types: lung squamous cell carcinoma [1], ovarian adenocarcinoma [11], hepatocellular carcinoma [44], gastric cancer [45], and colorectal cancer [46,47]. In ovarian cancer, high *SLC4A11* expression was associated with worse overall survival [11]. A comparison of gene expression data in normal ovarian epithelial tissue and primary serous ovarian cancer revealed that among 10 family members, *SLC4A11* expression was significantly upregulated in tumor tissue. In addition, it was shown that cases with lymphatic invasion had higher levels of *SLC4A11* expression than those without [11].

Mutations in the *SLC4A11* gene have also been shown to be associated with endometrioid adenocarcinoma [155]. Of the seven ovarian cancer biomarker genes considered, *SLC4A11* had the highest percentage of mutations. Bioinformatically, structural changes in the SLC4A11 protein have been shown to be associated with mutations, likely leading to decreased protein stability. However, the functional role of SLC4A11 in the development of endometrioid adenocarcinoma remains the subject of further research [155].

Increased expression of *SLC4A11* correlates with disease progression and poor prognosis in patients with gastric cancer [45,156]. There is a significant increase in *SLC4A11* gene and SLC4A11 protein levels in cancer tissues compared to normal tissues as confirmed by quantitative PCR, immunocytochemical analysis, and Western blot analysis. Bioinformatic analysis of data from the Oncomine database also confirmed the association between *SLC4A11* expression levels and clinical characteristics, revealing a significant association between high *SLC4A11* expression and poor clinical outcomes. These results indicate the potential of *SLC4A11* as a prognostic marker and therapeutic target for the treatment of gastric cancer [45,156]. In the context of tumor growth and metastasis, the mechanisms and functions of SLC4A11 remain unclear and require further investigation [46,47].

Evidence was obtained that the SLC4A11 protein functions as an NH_3_ transporter in HCC cells [157]. Despite its toxicity, NH_3_ can be utilized as a nitrogen source by tumor cells, activating lipid metabolism and supporting cancer stem cells (CSCs). Thus, SLC4A11 may mediate the mechanism of nitrogen metabolism rearrangement and promote tumor cell growth. In patients with cirrhosis, high NH_3_ levels correlate with increased HCC morbidity and poor survival prognosis. *SLC4A11* expression has been shown to be upregulated in CSCs. NH_4_Cl treatment of control hepatospheres from the human HCC cell line HepG2 and the mouse HCC cell line (mHCC) showed an increase in intracellular NH_3_ concentration, whereas *SLC4A11* KO cells showed no such effect. *SLC4A11* overexpression also contributed to the increase in NH_3_ levels in mHCC hepatospheres, as confirmed by quantitative PCR assessing the expression of *SLC4A11* and the CSC marker CD44. These results highlight the role of SLC4A11 as an NH_3_ importer in CSC in HCC [157].

However, SLC4A11 is unlikely to transport NH_3_ directly in the HCC cells. Recent works have shown that SLC4A11 transports H^+^ or OH^−^ (but not NH_3_/NH_4_^+^) across the membrane [18,57] This transport is activated by NH_3_ and by an increase in pH_i_ or pH_e_. In addition, no direct evidence for NH_3_ transport by the SLC4A11 protein has been obtained in HCC cells. Therefore, it is conceivable that NH_3_ accumulation in HCC cells may not be a result of direct NH_3_ transport by the SLC4A11 protein, but may have some other relationship with *SLC4A11* expression. For example, it could be a consequence of SLC4A11-mediated transport of H^+^/OH^−^ across the membrane.

NH_3_ can also accumulate in tumor cells as a result of glutamine catabolism. For example, in the corneal endothelium, which actively utilizes glutamine for energy production, this process was shown to be dependent on the presence of SLC4A11 [84]. Bonanno et al. suggested that the increased expression of *SLC4A11* in some types of cancer is associated with active glutaminolysis, “glutamine-addiction” [12]. Evidence has been obtained that glutamine is important for HCC cell proliferation as well as its use for energy needs [158].

SLC4A11 was reported to be located in the inner mitochondrial membrane [22]. Its function as an NH_3_-activated mitochondrial uncoupler involved in the reduction of oxidative stress was proposed [22]. While many studies support *SLC4A11*’s association with oxidative stress [80,81,82,83], a recent study found only weak co-localization of SLC4A11 with the mitochondrial marker COX-4 [36]. Instead, SLC4A11 and COX-4 co-localize mainly in structures resembling autophagosomes involved in mitochondrial recycling. This controversy calls for further investigation to clarify whether SLC4A11 truly resides in the inner mitochondrial membrane, which would confirm or refute its proposed role as a mitochondrial uncoupler. 

At the same time, cancer cells and corneal endothelium share another feature—lactate accumulation and its transport across the membrane. Cancer cells are characterized by the Warburg Effect—increased anaerobic respiration even at normal oxygen levels [12,159]. Rapidly proliferating cancer cells produce more acid equivalents through increased metabolism and intense glycolysis. However, they often maintain a pH_i_ equal to or more alkaline than normal cells, indicating active H^+^ removal. This may be accomplished by increasing the activity of acid extruders or decreasing the activity of acid-loading membrane transporters, as supported by data available in the literature [160,161]. The role of some members of the SLC4 family has been demonstrated in several types of cancer that regulate tumor cell pH_i_ through ion transport [162]. We have discussed above the link between SLC4A11 and lactate efflux in corneal endothelium (see Section 3.1.4. for details). Thus, SLC4A11 may also be involved in the process of tumor cell pH regulation, but this mechanism requires further investigation.

*SLC4A11* is increasingly recognized as a prognostic marker due to its elevated expression and certain mutations in various cancers. SLC4A11 may support tumor progression through metabolic adaptation and mitochondrial and pH regulation. However, important questions remain unanswered. The precise substrates transported by SLC4A11 in cancer cells, whether NH_3_ directly or primarily H^+^/OH^−^ ions activated by NH_3_, are not fully understood. The localization of SLC4A11 in mitochondria needs to be further elucidated. Additionally, the function of SLC4A11 in glutamine-addicted tumors is only beginning to be elucidated.

## 5. Conclusions

The *SLC4A11* gene, which encodes a unique membrane transporter with tissue-specific expression, has emerged as a critical player in the physiology and pathology of the corneal endothelium and other tissues. Over the past two decades, research has vastly expanded our understanding of SLC4A11, revealing a complex landscape of transcript variants, protein isoforms, and diverse roles in cellular functions. A number of publications report ubiquitous expression of *SLC4A11*, while in other studies *SLC4A11* transcription is not detected in some tissues. To date, the expression of three transcript variants of *SLC4A11* (transcripts 1–3) and their corresponding protein isoforms has been characterized. Further research is required to elucidate the expression profile of *SLC4A11*, particularly in light of the identification of novel transcript variants and the cellular heterogeneity present within tissues. As a result of extensive study, researchers have reached limited consensus on SLC4A11’s transport function. Current evidence identifies it as an NH_3_- and pH-regulated H^+^/OH^−^ transporter. Additionally, SLC4A11 has been implicated in lactate transport, though the proposed mechanism remains unconfirmed. The exact relationship between MCT4 (a lactate transporter) and SLC4A11 also requires clarification. Emerging data also highlight SLC4A11’s potential function as an adhesion molecule and its influence on cell fate determination pathways, both of which warrant further investigation. SLC4A11’s involvement in pH regulation, lactate:H^+^ transport, glutaminolysis, and overexpression in certain cancers opens new avenues for exploring its function in glutamine-addicted cancers. Studies have suggested that SLC4A11 acts as a mitochondrial uncoupler, but recent findings contradict its localization to mitochondrial membranes. Instead, they emphasize the need to investigate its role in autophagy and mitophagy processes. While *SLC4A11* pathogenic variants are firmly linked to CHED and Harboyan syndrome, its causal role in FECD lacks sufficient evidence. While much has been learned, the field is poised for further breakthroughs that will clarify the biological significance of SLC4A11 isoforms, resolve current controversies, and ultimately translate into therapies for patients affected by SLC4A11-related disorders.

## Figures and Tables

**Figure 2 biomolecules-15-00875-f002:**
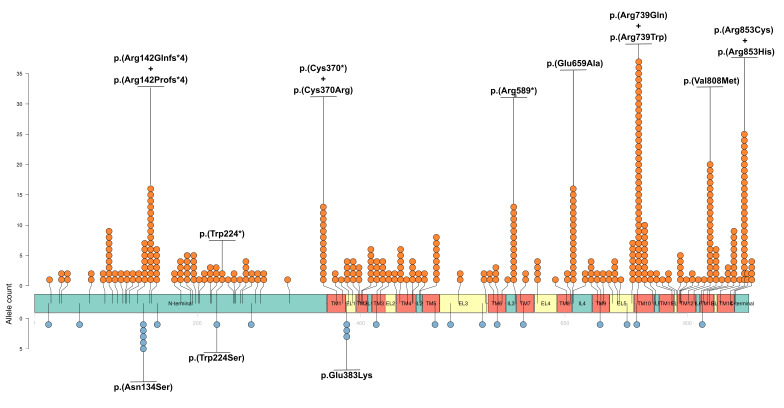
Distribution of *SLC4A11* variants reported in CHED/Harboyan syndrome and FECD cases across isoform v3 (NP_001167560.1). Protein domains (*x*-axis) are annotated based on UniProt (Q8NBS3-3) [142]. The *y*-axis represents allele counts aggregated from Appendix A, with counts for variants affecting identical residues summed, excluding splice-site variants. Cytoplasmic domains (green), transmembrane regions (TMs, red), and extracellular domains (yellow) are color-coded. CHED/Harboyan syndrome-associated variants are highlighted in orange, and FECD-associated variants in blue.

**Table 2 biomolecules-15-00875-t002:** SLC4A11-mediated NH_3_/NH_4_^+^ and H^+^/OH^−^ transport mechanisms proposed in publications.

NH_3_/NH_4_^+^ Transport	H^+^/OH^−^/H_2_O Transport	Reference
mediates NH_4_^+^ transport into the cell	Na⁺ enters the cell, H⁺ leaves the cell, or cotransport of Na⁺ and OH⁻ into the cell together	[16,17]
carries out cotransport of NH_3_ and 2H^+^ (possibly in both directions)		[20] [53]
mediates NH_3_ transport into the cell	does not carry out additional transport of any ions	[19,25]
	transport of H^+^/OH^−^ regardless of the presence of NH_3_/NH_4_^+^	[18]
carries out electrogenic NH_3_ -H^+^ cotransport	transport of H^+^ in both Na^+^-independent and Na^+^-bound modes; water transport in the presence of an osmotic gradient	[15]
two competing membrane processes: H^+^/OH^−^ conductance and cotransport of H^+^ with NH_3_ or as part of NH_4_^+^ ions		[21]
	extracellular pHₑ-dependent H⁺/OH^−^ transport	[54,55,56]
NH_3_/NH_4_^+^ act as allosteric activators of SLC4A11, influencing H^+^/OH^−^ conductance		[57]

## Data Availability

No new data were created or analyzed in this study.

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
