# Peer review of "SLC4A11 Revisited: Isoforms, Expression, Functions, and Unresolved Questions"

_biomolecules, 2025, doi:10.3390/biom15060875_

Round 1
Reviewer 1 Report
Comments and Suggestions for Authors This review explores the diversity of SLC4A11 transcripts and isoforms, as well as how this complexity affects the interpretation of their tissue expression patterns and their functional roles in health and disease. It provides a comprehensive update on SLC4A11 biology and identifies key gaps for future research. However, this article still has some flaws:1. The RefSeq database lists 20 transcription variants, and the article mainly describes transcription variants 1-3. Do other transcription variants have potential functions;
2. The expression in Figure 2 is not intuitive, and the caption is also very brief. Regarding the transport function of SLC4A11 (whether it transports Na+, NH3, etc.), it is recommended to create a table that is more intuitive;
The title and content of "4.1" do not seem to match. The title seems to confirm the impact of SLC4A11 variant on FECD, while the content revolves around "controversy";
4. Incorrect numbering in Part 3;
5. As a review, there are few figures;
6. Inconsistent reference format;
Author Response
Comment 1: The RefSeq database lists 20 transcription variants, and the article mainly describes transcription variants 1-3. Do other transcription variants have potential functions;
Response 1:
We appreciate this insightful remark. In order to facilitate a discussion of the potential roles of transcript variants and isoforms other than 1-3, the revised manuscript has been updated with additional information regarding their structural characteristics and potential functional roles (Lines 133-162 ). Also, we provide the added text below:
“All protein-coding transcript variants of SLC4A11 listed in RefSeq, which include sequences encoding the full-length transmembrane domain, correspond to protein isoforms SLC4A11-v1, SLC4A11-v2/SLC4A11-v2M36, SLC4A11-v2M36, and SLC4A11-v3. However, RefSeq also contains transcripts that lack the exon corresponding to exon 10 of NM_032034 (e.g., NM_001363745, NM_001400280, XM_047440543), which are the result of alternative splicing. Exon 10 encodes the second transmembrane region and most of the third transmembrane region. Its absence produces protein products missing these segments.
Additionally, some predicted transcripts include truncated exons 13 and 15 (numbered according to NM_032034). These alterations affect extracellular loops that are part of the transmembrane domain. Truncation of exon 13 removes part of the large extracellular loop 3 (EL3), which contains N-glycosylation sites. However, the sites themselves remain intact. Truncated exon 15 results in the loss of most of extracellular loop 4 (EL4). It is noteworthy that these splicing events do not result in frameshifts, indicating that the resulting proteins are canonical isoforms with deletions in their polypeptide chains. The inclusion of truncated exons 13 and 15 does not disrupt transmembrane regions but shortens intervening polypeptide segments. Deletions in EL3 and EL4 may either preserve normal protein function or severely compromise it, particularly if such deletions affect the structural organisation of the transmembrane domain (e.g., proper folding or spatial orientation of transmembrane regions).
RefSeq also lists potential transcripts encoding C-terminally truncated SLC4A11 isoforms, a consequence of premature transcription termination. These variants would lack the C-terminal domain and part of the transmembrane domain (up to all transmembrane regions except the first). Such isoforms could differ markedly from canonical full-length transmembrane domain isoforms in subcellular localization, maturation, and function.
The functional role of SLC4A11 isoforms with polypeptide deletions or truncations (via alternative splicing or premature termination) remains unexplored. Priority should be given to analyzing the prevalence of these splice variants. They may lack functional significance, representing low-abundance stochastic splicing byproducts. However, if produced at substantial levels, functional assays could elucidate their cellular roles.”
Comment 2: The expression in Figure 2 is not intuitive, and the caption is also very brief. Regarding the transport function of SLC4A11 (whether it transports Na+, NH3, etc.), it is recommended to create a table that is more intuitive;
Response 2: We agree that Figure 2 was overly simplistic. Our goal was to provide a basic overview of the potential transport functions of SLC4A11. We created Table 2, which contains more information and is more intuitive (Lines 343-345).
Comment 3: The title and content of "4.1" do not seem to match. The title seems to confirm the impact of SLC4A11 variant on FECD, while the content revolves around "controversy";
Response 3: We greatly appreciate your suggestion to make our manuscript clearer. We changed the title of the Chapter 4.2 (former 4.1) to “Limited evidence for a causal role of SLC4A11 variants in FECD”.
Comment 4: Incorrect numbering in Part 3;
Response 4: We thank the reviewer for noticing the error in numbering. We have corrected the numbering in Chapter 3.
Comment 5: As a review, there are few figures;
Response 5: We have added the new Figure 2 in Chapter 4 (Line 775) describing the location and prevalence of variants in the SLC4A11 gene in CHED and FECD cases.
Comment 6: Inconsistent reference format;
Response 6: We have taken the time to review the references once more and fixed the errors in Lines 38, 40, 111, 166 (Figure 1 caption).
Reviewer 2 Report
Comments and Suggestions for Authors
The paper "SLC4A11 Revisited: Isoforms, Expression, Functions and Unresolved Questions" by Kovaleva et al, is a comprehensive review on Slc4a11 function, and proposed roles in corneal endothelial dystrophy and cancer. The paper is well written and organized and could be a valuable resource for investigators working in the function of this protein as it offers interesting insights and new hypothesis.
I recommend: Accept after minor revisions
Revisions:
Line 40: Change "a causal role in FECD remains insufficient [10." for "a causal role in FECD remains insufficient [10]"
Lines 280-282: Replace the sentence "The importance of the 280 cytoplasmic domain, especially the first 300 amino acid residues, for SLC4A11 dimerization [50, 51]" for "he importance of the 280 cytoplasmic domain, especially the first 300 amino acid residues, for SLC4A11 dimerization has been stablished [50, 51]"
Line 302: Change "trantpor for "transport".
Lines 483-485. "Decreased SLC4A11 expression leads to lower NRF2 and downstream antioxidant gene expression, impairing the cell’s ability to counteract oxidative stress and resulting in increased ROS and cytotoxicity". Slc4a11 has an element in its promoter that can explain regulation of Slc4a11 by Nrf2. How Slc4a11 regulates Nrf2 expression or translocation to the nucleus? Is there any hypothesis in the literature?
Line 486-488. "SLC4A11 is also crucial for maintaining mitochondrial integrity. Loss of SLC4A11 has been shown to cause mitochondrial membrane depolarization, an early event in apoptosis". Provide a reference.
Line 873-874. Change "The evidence was obtained that the SLC4A11 protein functions as an NH3 transporter in HCC cells [126]." for "Evidence was obtained that the SLC4A11 protein functions as an NH3 transporter in HCC cells [126]."
Author Response
Comment 1: Line 40: Change "a causal role in FECD remains insufficient [10." for "a causal role in FECD remains insufficient [10]"
Response 1: We thank the reviewer for pointing out the error in the reference. We have corrected it (Line 40).
Comment 2: Lines 280-282: Replace the sentence "The importance of the 280 cytoplasmic domain, especially the first 300 amino acid residues, for SLC4A11 dimerization [50, 51]" for "the importance of the 280 cytoplasmic domain, especially the first 300 amino acid residues, for SLC4A11 dimerization has been established [50, 51]"
Response 2: We would like to express our gratitude to the reviewer for bringing the error in this sentence to our attention. We have corrected it (Line 312).
Comment 3: Line 302: Change "transpor for "transport".
Response 3: We acknowledge the error in the word and have taken action to rectify it (Line 332 of the revised manuscript).
Comment 4: Lines 483-485. "Decreased SLC4A11 expression leads to lower NRF2 and downstream antioxidant gene expression, impairing the cell’s ability to counteract oxidative stress and resulting in increased ROS and cytotoxicity". Slc4a11 has an element in its promoter that can explain regulation of Slc4a11 by Nrf2. How Slc4a11 regulates Nrf2 expression or translocation to the nucleus? Is there any hypothesis in the literature?
Response 4: We would like to thank the reviewer for the insightful question. We will discuss it in more detail below. The extant literature contains only a limited number of hypotheses regarding this relationship (Lovatt, Matthew, et al. "Nrf2: A unifying transcription factor in the pathogenesis of Fuchs’ endothelial corneal dystrophy." Redox Biology 37 (2020): 101763.). The precise mechanism by which SLC4A11 regulates antioxidant signalling remains an area of future research and is not yet fully understood. The hypothesis that a positive regulatory feedback loop exists between SLC4A11 and NRF2, with the potential to mobilise the cellular antioxidant machinery, is presented. SLC4A11 has been hypothesised to regulate optimal NRF2 activation in corneal endothelial cells through direct interaction or by promoting protein stabilisation (Guha, Sanjukta, et al. "SLC4A11 depletion impairs NRF2 mediated antioxidant signaling and increases reactive oxygen species in human corneal endothelial cells during oxidative stress." Scientific reports 7.1 (2017): 4074.).
We added the following sentence to lines 515-517: "The precise mechanism by which SLC4A11 regulates antioxidant signaling remains an area of future research and is not yet fully understood."
Comment 5: Line 486-488. "SLC4A11 is also crucial for maintaining mitochondrial integrity. Loss of SLC4A11 has been shown to cause mitochondrial membrane depolarization, an early event in apoptosis". Provide a reference.
Response 5: We thank the reviewer for pointing out this omission. We have added a reference to the sentence in question in Line 520: Ogando, Diego G., et al. "Ammonia sensitive SLC4A11 mitochondrial uncoupling reduces glutamine induced oxidative stress." Redox biology 26 (2019): 101260.
Comment 6: Line 873-874. Change "The evidence was obtained that the SLC4A11 protein functions as an NH3 transporter in HCC cells [126]." for "Evidence was obtained that the SLC4A11 protein functions as an NH3 transporter in HCC cells [126]."
Response 6: We corrected this sentence (Lines 977-978).
Round 2
Reviewer 1 Report
Comments and Suggestions for Authors
The author answered all my questions and made revisions in the revised manuscript. The author's response is reasonable, and the author's revisions to the article are appropriate. I am satisfied with the work done by the author.